# Attitudes of medical students on conflict of interest: A comparative study of Korea and France

Hoseob Ji[1], Byung-in Choe[2]*

**1** Medical Training Team, Hallym University Sacred Heart Hospital, Anyang-si, South Korea, **2** Nicholas Cardinal Cheong Graduate School for Life, The Catholic University of Korea, Seoul, South Korea

* bichoe@gmail.com

## Abstract

Medical students are potential marketing targets for pharmaceutical companies because established prescribing habits are not easily altered. In 2014, Bruno Etain and several other researchers published a paper which investigated the knowledge of and opinions on potential conflict of interest (COI) with regard to preclinical and clinical students enrolled in medical schools in France and residents working in hospitals. An empirical survey study with Korean medical students concerning their educational experiences and views on conflicts of interest and comparing and contrasting the results with Etain's study of French medical students. Receipt of direct or indirect financial offerings from pharmaceutical industries was not properly recognised as COI by the medical students. Therefore, strengthening education on COI and implementing institutional improvements for COI disclosure are essential to prevent bias caused by COI and enhance awareness levels regarding COI.

## Introduction

There are many studies [1, 2] demonstrating the impact of pharmaceutical companies' marketing activities on doctors' prescribing practices. For these reasons, pharmaceutical companies are conducting various direct and indirect marketing activities to influence doctors' prescriptions.

Medical students, interns and residents (hereafter referred to as "students") are potential marketing targets for pharmaceutical companies because established prescribing habits are not easily altered [3].

A conflict of interest (hereafter referred to as "COI") is a set of circumstances that creates a risk that professional judgment or actions regarding a primary interest(ex. the welfare of patients) will be unduly influenced by a secondary interest(ex. financial gain) [4].

COI is a specific 'circumstance.' Therefore, while it may not be a problem in itself, it has a high likelihood of leading to biases. Hence, proactive measures are necessary to prevent it, and "disclosure" is widely recognized as a common solution [5].

Recently in Korea, an Act on the Prevention of COI Related to Duties of Public Servants was passed on May 18, 2021 with its main content featuring the prohibition of public officials

**Data Availability Statement:** All relevant data are within the manuscript and its Supporting information files.

**Funding:** The author(s) received no specific funding for this work.

**Competing interests:** The authors have declared that no competing interests exist.

from pursuing personal interests in their duties [6]. In addition, the Pharmaceutical Affairs Act was revised on July 20, 2021 to prevent rebates in pharmaceutical sales.

In general, rebates refer to the act or the amount by which a seller reduces the selling price by refunding a portion of the sale to the buyer to promote sales.

However, under the Korean Pharmaceutical Affairs Act, the pharmaceutical sales rebate refers to a 'kickback' wherein a doctor receives private profits in correspondence to the prescription and purchase of necessary drugs from a pharmaceutical company or supplier while prescribing drugs for medical treatment.

Types of pharmaceutical sales rebates announced by Korea's Health Insurance Review and Assessment Service (HIRA) include ① drug adoption case fees, ② drug prescription case fees, ③ regular advisory fees, ④ tourism or meal expenses, and ⑤ expenses related to overseas conference attendance.

In particular, the revised Pharmaceutical Affairs Act mandates the preparation, storage, submission, and disclosure of an expenditure report on the provision of economic benefits by pharmaceutical suppliers. Moreover, it grants the Minister of Health and Welfare the authority to investigate the actual condition of the expenditure report and announce the results. In addition, it stipulates the obligations related to prohibiting the provision of rebates and submitting expenditure reports of pharmaceutical sales representatives.

According to the Ministry of Health and Welfare's 'Notification Status of Illegal Rebate Detection between 2015 and 2018,' it appears that illegal rebates have been gradually decreasing. However, it has been discovered that rebates are not actually decreasing, but rather that pharmaceutical companies are providing rebates through sales agencies known as CSOs (Contract Sales Organizations), resulting in their exclusion from the statistics. They are taking advantage of the fact that CSOs are not classified as drug suppliers under the Pharmaceutical Affairs Act, making it impossible to impose penalties even if they are caught [7].

In 2014, Bruno Etain and several other researchers published a paper which investigated the knowledge and opinions regarding COI in relation to preclinical and clinical students enrolled in medical schools in France as well as residents working in hospitals.(8) In this paper, many of the respondents reported that they thought they knew about COI but did not properly identify it in its proper context; as such, they reported they were not properly educated on COI. In addition, many students were exposed to pharmaceutical representatives, and they held the contradictory belief that biases could affect others but not themselves.

In Korea, as in France, all students are frequently exposed to marketing by pharmaceutical sales representatives and are likely to face various situations involving a COI.

In this context, we also conducted a survey of students in Korea to collect quantitative information on knowledge, education, personal exposure, and opinions about COI and to identify priorities for future education on COI. In addition, we tried to compare the differences in attitudes toward COI between Korean and French students.

This paper aims to identify the following: ① Korean students' ability to identify some common situations as a potential COI; ② Korean students reports and views about their education on COI; ③ Korean students perceptions of the bias induced by COI on themselves and others.

## Methods

Prior to the start of the study, we asked Etain, the author of the previous research paper [8], if it would be acceptable to conduct a study applying the same research contents to Korean students. Subsequently, we received his consent for the study.

Next, the questionnaire used in the previous study was received and translated. In order to reduce translation errors, a questionnaire was distributed to 10 students in advance to evaluate their understanding and reflect their opinion on the subject.

After receiving IRB approval in January 2022, we called the administrative department of a medical institution designated as a medical school and resident training institution in Korea. The purpose of the survey was explained, and a request was made to collect the survey; accordingly, a written questionnaire was sent to organizations that accepted the request. A link to the online survey was sent to the residents of the institution where the author is working. The survey ended in June 2022, and data extraction and analysis were carried out.

Participation was voluntary, anonymous, and open to all students. This involved those who were considered preclinical (first two years of medical school); clinical (three to six years, intern); and residents (one to four years). The potential participants were students who received questionnaires through staff at the institution, but the exact number of students who received them was unknown.

All statistical analyses were conducted using the 'statsmodels' and 'scipy' packages in Python; the same analysis method was used for comparative analysis with the previous research paper. We were able to obtain the mean and standard deviation (SD) for continuous variables. Chi-square tests were used for the comparison of categorical variables. Logistic regression analysis was performed to assess the effect of selected variables on the perceived potential influence of COI on prescriptions (for self and for others). The results were checked using beta, standard deviation, p-value, and the odds ratio. A P-value of less than 0.05 was considered statistically significant.

## Results

In total, 388 students participated (56 preclinical [PC], 251 clinical [C], and 81 residents [R]). A large number of students reported insufficient education on COI (63.7%) and reported infrequent attendance in lectures (8.5%) or personal research (10.3%). Additionally, over half of the respondents (54.6%) expressed interest in COI information regarding their courses.

Upon comparing the results of previous studies conducted in France and Korea, the proportions of the two items(3.3./3.4.) were similar, as depicted in the Table 1 below. However, it was notable that French students expressed a significantly higher level of dissatisfaction regarding the adequacy of COI information(3.2./4.6.).

In response to the question, "Do you think you know how to define what a COI is?", only 24.8% responded that they could define it, and there was no significant difference by class level (PC: 26.8%; C: 25.9%; R: 32.1%; P = 0.1385). Compared to the results of a previous study in

**Table 1.** Medical students' perceptions and experiences regarding COI information.

| Question | (% of students) | | |
|---|---|---|---|
| | F | K | Difference |
| 3.2. During your studies, do you feel that you received enough information about declaration of COI? (responding "No") | 85.2 | 63.7 | -21.5 |
| 3.3. During your studies, did you receive a lecture or a tutorial on COI? (responding "Yes") | 4.3 | 8.5 | 4.2 |
| 3.4. During your studies, did you do any personal research (internet for example) on the impact of COI or on the declaration of COI? (responding "Yes") | 11.1 | 10.3 | -0.8 |
| 4.6. I would like to know the COI of my teachers when they teach me? (responding "Yes") | 66.6 | 54.6 | -12.0 |
| 4.5. Do your teachers mention their COI during their lessons? (responding "No") | 66.5 | 46.4 | -20.1 |

* COI: conflict of interest * F: French; K: Korea

which (64.6%) of French students claimed they could define it, Korean students' knowledge about COI was found to be low (24.8%).

In addition, Students who answered "Yes" to at least one of the following two questions were 51 (13.1%): "Do you think you received enough information about COI disclosure?" and "Have you ever received a special lecture or individual training on COI?". Furthermore, out of the 51 respondents who answered "Yes" to at least one of the questions, 23 (45.1%) of them responded with "No" or "I'm not sure" to the question "Do you think you know how to define what a COI is?", indicating that COI education is not being properly conducted.

Table 2 shows the degree of awareness of students about COI risk. "Set 1" contains questions about situations involving indirect financial offerings (mainly gifts, meals, training, or meetings). In comparison, "Set 2" concerns questions about situations involving direct financial offerings (including research participation, payment of lecture fees and stock holdings).

Give this, 35.5% in Set 1 and 45.6% in Set 2 considered monetary offerings as COI, demonstrating a higher awareness of direct offerings than indirect offerings. However, the percentage was still at a low level, occupying less than half of the set. With the exception of a close relative employed by the PI, we observed a moderately increased recognition of COI for all situations as the class level increased. P-values of less than 0.05 were considered statistically significant when comparing the responses of students according to class level.

Compared to the results of previous studies in France, Set 1 (34.5%) was at a similar level to Korea, but Set 2 (71.7%) showed a significantly higher level of awareness than Korean students. Among them, the largest difference was seen for holding stock, and the lowest difference was for receiving small gifts such as books and pens. In addition, French students showed similar COI recognition rates by class level, but Korean students showed a large difference.

In the additional analysis of Table 2, we applied the responses to the question "Do you think you can define what a conflict of interest is?" instead of the responses to the question "What is your class level?" The results showed that respondents who answered "Yes" had a higher awareness of COI (50.1%) compared to those who answered "No" (31.8%). Furthermore, among the

**Table 2. Medical students' knowledge of situations at risk of COI.**

| Question or Statement | What is your class level? | | | | | | | | | |
|---|---|---|---|---|---|---|---|---|---|---|
| Do you consider the following situation as a COI? (% of students responding "Yes") | All | | Preclinical | | Clinical | | Residents | | P-value | |
| | F | K | F | K | F | K | F | K | F | K |
| Set 1 | | | | | | | | | | |
| Receiving a gift of minor value (e.g., book, pen) from the PI | 27.7 | 30.9 | 26.5 | 26.8 | 27.0 | 28.7 | 36.2 | 40.7 | < .0001 | .0011 |
| Having a close relative employed by the PI | 32.9 | 39.7 | 30.5 | 14.3 | 37.4 | 45.4 | 42.3 | 39.5 | < .0001 | < .0001 |
| Being invited for lunch/dinner by the PI | 35.0 | 45.1 | 41.1 | 26.8 | 35.9 | 46.6 | 38.0 | 53.1 | .0001 | < .0001 |
| Participating in a training sponsored by the PI | 35.6 | 26.5 | 35.4 | 21.4 | 40.3 | 24.3 | 40.8 | 37.0 | .0002 | < .0001 |
| Being invited to a conference by the PI | 41.5 | 35.1 | 46.5 | 14.3 | 41.9 | 36.7 | 49.4 | 44.4 | .0002 | < .0001 |
| Set 2 | | | | | | | | | | |
| Participating in a clinical study paid by the PI | 56.0 | 43.8 | 50.1 | 30.4 | 58.9 | 42.6 | 73.7 | 56.8 | < .0001 | .0003 |
| Receiving a fellowship from the PI | 58.7 | 49.2 | 55.4 | 35.7 | 66.7 | 47.0 | 72.6 | 65.4 | < .0001 | < .0001 |
| Being paid as a speaker by the PI | 69.2 | 25.8 | 71.2 | 17.9 | 78.2 | 24.3 | 76.3 | 35.8 | < .0001 | < .0001 |
| Holding stock shares in the PI | 85.5 | 41.8 | 84.3 | 16.1 | 92.7 | 45.8 | 90.7 | 46.9 | < .0001 | < .0001 |
| Receiving a salary or honoraria from the PI | 89.1 | 67.3 | 85.0 | 25.0 | 95.1 | 71.3 | 96.6 | 84.0 | < .0001 | < .0001 |

COI: Conflict of interest; PI: Pharmaceutical industry; F: French; K: Korea

* p-values were obtained from Chi-square tests for homogeneity between preclinical students, clinical students, and residents.

**Table 3. Exposure to marketing strategies, potential consequences of COI for self and others, and transparency.**

| Question or Statement (% of students responding "Yes") | What is your class level? | | | | | | | | | |
|---|---|---|---|---|---|---|---|---|---|---|
| | All | | Preclinical | | Clinical | | Residents | | P-value | |
| | F | K | F | K | F | K | F | K | F | K |
| Exposure to marketing strategies | | | | | | | | | | |
| Have you ever met a representative of the PI? | 63.9 | 39.9 | 18.2 | 5.4 | 79.4 | 37.5 | 96.6 | 71.6 | < .0001 | < .0001 |
| Have you ever received a gift from the PI? | 62.7 | 15.7 | 28.1 | 3.6 | 71.8 | 8.8 | 89.9 | 45.7 | < .0001 | < .0001 |
| Consequences of COI for others | | | | | | | | | | |
| COI can induce bias in medical training | 64.5 | 54.1 | 65.8 | 62.5 | 61.8 | 54.2 | 66.5 | 48.1 | .13 | .2531 |
| COI can induce bias in drug prescriptions | 87.9 | 75.8 | 89.3 | 55.4 | 89.7 | 79.3 | 84.1 | 79.0 | .003 | .0005 |
| COI can induce bias in research | 86.6 | 79.6 | 82.3 | 75.0 | 90.1 | 80.1 | 87.0 | 81.5 | < .0001 | .6241 |
| Self-consequences of COI | | | | | | | | | | |
| Having received a gift will influence your future prescriptions | 2.4 | 8.2 | 1.5 | 8.9 | 2.0 | 8.4 | 3.7 | 7.4 | .27 | .2002 |
| I consider it as a COI when attending a meal sponsored by the PI | 21.4 | 17.3 | 10.8 | 10.7 | 24.4 | 15.5 | 29.5 | 27.2 | < .0001 | .0003 |
| Transparency | | | | | | | | | | |
| Patients should be informed of their physicians' COI | 39.4 | 25.5 | 43.9 | 25.0 | 40.3 | 30.3 | 33.3 | 11.1 | .002 | < .0001 |
| I favor a public declaration of COI (e.g., Ministry of Health website) | 65.0 | 47.2 | 61.9 | 46.4 | 67.8 | 52.2 | 64.9 | 32.1 | .08 | < .0001 |

COI: Conflict of interest; PI: Pharmaceutical industry; F: French; K: Korea

* p-values were obtained from Chi-square tests for homogeneity between preclinical students, clinical students and residents.

respondents who answered "Yes," the awareness of direct COI (Set 2) (56.0%) was higher than the awareness of indirect COI (Set 1) (44.1%).

Table 3 shows the degree of exposure to COI, the degree of influence of COI, and thoughts on the disclosure of COI. As the students' education levels go up, they are more exposed to pharmaceutical industry representatives and receive more gifts.

Regarding the question of whether a COI can cause bias, the percentage of those who think others can develop a bias is high (69.8%), while the percentage of those who think they can develop a bias is low (12.8%), thereby showing a contradictory tendency. This was similar to the French students in previous studies, but we found that the French students had a greater gap regarding the likelihood of bias occurring in others (79.7%) and in themselves (11.9%).

Table 4 demonstrates a 95% confidence interval indicating that for each increment in the class level, the likelihood of answering 'Yes' to that question increases by a minimum of 1.17 times and a maximum of 2.65 times.

When asked about transparency in disclosing their COI to patients or the public, residents in the higher education level were less likely to disclose than preclinical/clinical students, showing similar educational-level tendencies to French students, but with lower response rates.

**Table 4. Medical students' perceptions of bias by class level.**

| Question | Class level (preclinical→clinical→resident) | |
|---|---|---|
| | p-value | OR 95% CI |
| Do you think COI can induce bias in drugs prescription for others? | 0.007 | 1.76 (1.17~2.65) |

COI: Conflict of interest

The reference category was the group of students answering "no" to the questions.

Note: The analyses were performed using gender as a covariate.

In particular Table 3, the responses to the question "Have you ever received a gift from the PI?" showed the largest difference between Korean (15.7%) and French (62.7%) students, while the responses to the question "I consider it as a COI when attending a meal sponsored by the PI" showed the smallest difference between Korean (17.3%) and French (21.4%) students.

Further analysis of Table 3, we also examined the responses to the question "Do you think you can define what a conflict of interest is?" instead of the question "What is your class level?" As a result, the overall percentage of respondents who answered "Yes" (42.7%) and "No" (38.4%) was similar. Furthermore, among those who answered "Yes," the percentage of those who believe that others can develop bias (68.2%) was higher than the percentage of those who believe they can develop bias (14.2%).

## Discussion

Compared to the results of the survey in France, our survey shows similar tendencies overall. This was a similar tendency to be observed in the United States, Canada, and other countries. [9, 10] Additionally, while each study conducted in France and Korea was not specifically designed to demonstrate differences, it would be meaningful to compare the research findings between the two countries using a consistent approach to validate any potential disparities.

This study showed that Korean students could not properly identify COI situations. As their education level goes up, they are more exposed to pharmaceutical industry representatives and receive more gifts. In addition, Korean students presented a contradictory idea that bias might happen to other people, but bias would not happen to them similar to the results in previous study.

Further analysis revealed that respondents who believed they had knowledge about COI (18.9%) reported a higher incidence of receiving gifts compared to those who believed they lacked knowledge (3.0%). On the other hand, these results suggest that students with more exposure to pharmaceutical salespeople or receiving gifts had a better understanding of COI. Therefore, further analysis through follow-up studies is necessary.

The strengths of this study include the diverse participation of medical students, interns, and residents at all stages of medical education. In particular, residents evenly participated in 18 out of the 26 training departments operating in Korea, with the exception of 5 departments with a total quota of less than 50.

However, The main potential limitation of this study is the lack of representation, which is also showed in previous studies. The majority of respondents were locally concentrated in some regions (Seoul, Gyeonggi, Gangwon), with more than half of them being enrolled at the clinical level (PC: 14.4%; C: 64.6%; R: 20.8%). This imbalance is expected to reflect the following reasons: The survey on COI for students in Korea is the first attempt. In addition, many surveyed organizations were reluctant to distribute the questionnaire due to lack of understanding of COI and administrative burden.

The lack of representation is thought to be partially offset by the following points: (1) This is the first attempt in Korea to investigate the COI awareness of students, so it could provide an important basis on the need for COI education in future medical education; and (2) it provides quantitative data on potential differences in perception according to curriculum stage, including among preclinical and clinical students.

Interestingly, the "Yes" response was high (69.8%) in the "Consequences of COI for others" question, while the "Yes" response was low (12.8%) in the "Self-consequences of COI" question. This can be interpreted as a contradictory situation, but if you think about it differently, it can be interpreted that many students have a basic sense of ethics to prevent bias from occurring due to COI.

Currently, medical ethics education for medical students in Korea faces challenges due to an absolute lack of class time, insufficient presence of related scholars, unclear educational goals, and inadequate development of methodologies. Additionally, there is a lack of education specifically addressing COI [11, 12].

Therefore, if students spend more time on quality COI education to clearly understand the concept of COI and improve their ability to judge COI situations accurately and develop individual ethical awareness, then this will be the best way to prevent bias from occurring.

Next, there will have to be systematic support. This should involve your own COI disclosure method as well as a convenient and diversified method of requesting others' COI disclosures to prevent the occurrence of bias caused by COI and increase the level of awareness.

## Supporting information

**S1 File.**
(XLSX)

## Acknowledgments

We acknowledge the participants who agreed to the survey and the institution who agreed to distribute the questionnaire. We particularly acknowledge Daehyeon Bae(School of Cybersecurity, Korea University) for his help with statistical analysis.

## Author Contributions

**Conceptualization:** Hoseob Ji, Byung-in Choe.

**Data curation:** Hoseob Ji.

**Formal analysis:** Hoseob Ji.

**Investigation:** Hoseob Ji.

**Project administration:** Hoseob Ji.

**Supervision:** Byung-in Choe.

**Validation:** Byung-in Choe.

**Visualization:** Hoseob Ji.

**Writing – original draft:** Hoseob Ji.

**Writing – review & editing:** Byung-in Choe.

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
