## [Decision Letter · Decision Letter 0]

6 Jun 2023

PONE-D-23-08482Attitudes of medical students on conflict of interest: A comparative study of Korea and FrancePLOS ONE

Dear Dr. Choe,

Thank you for submitting your manuscript to PLOS ONE. After careful consideration, we feel that it has merit but does not fully meet PLOS ONE’s publication criteria as it currently stands. Therefore, we invite you to submit a revised version of the manuscript that addresses the points raised during the review process.

The two reviewers have provided many useful and constructive comments and suggestions for improvement. Please take them into account when revising your manuscript or explain your reasons not to do so.

We look forward to receiving your revised manuscript.

Kind regards,

Alberto Molina Pérez, Ph.D.

Academic Editor

PLOS ONE

Journal Requirements:

2. Please provide additional details regarding participant consent. In the ethics statement in the Methods and online submission information, please ensure that you have specified what type you obtained (for instance, written or verbal, and if verbal, how it was documented and witnessed). If your study included minors, state whether you obtained consent from parents or guardians. If the need for consent was waived by the ethics committee, please include this information

Reviewers' comments:

Reviewer's Responses to Questions

**Comments to the Author**

1. Is the manuscript technically sound, and do the data support the conclusions?

Reviewer #1: Yes

Reviewer #2: Partly

2. Has the statistical analysis been performed appropriately and rigorously? 

Reviewer #1: Yes

Reviewer #2: No

3. Have the authors made all data underlying the findings in their manuscript fully available?

Reviewer #1: Yes

Reviewer #2: Yes

4. Is the manuscript presented in an intelligible fashion and written in standard English?

Reviewer #1: Yes

Reviewer #2: No

5. Review Comments to the Author

Reviewer #1: Thank you very much for your study

I have some comments that I list in chronological order

1. Line 27: “lecture participation in COI affected future prescriptions more” is unclear

2. Line 50: “he was exposed to pharmaceutical sales representatives a lot, and he had a contradictory idea that he would not be biased, although others could be” is unclear: one could understand that Etain himself is exposed to pharmaceutical sales representatives while its conclusion concerns students.

3. Line 70: “a questionnaire was distributed to 10 students in advance to evaluate their 70 understanding and reflect their opinion on the subject”. Did you modify the questionnaire according to this first evaluation?

4. Line 79: “The potential participants » : can you precise how many potentials participants were expected?

5. Line 93: “Although they expressed interest in whether the courses they attended concerned COI (54.6%), about half said the professors teaching the courses did not disclose their COI information (46.4%)”: I think that these ideas are not related and should not be in the same sentence.

6. Lines 105 to 107: the 51 students are probably the same in the 2 part of the paragraph but can you precise?

7. Table 1: can you precise that Etain and al wanted to investigate the Medical Students’ Knowledge of Situations at Risk of COI according to their level within medical school (and so did you)?

8. Line 120 and 121: can you precise what this percentages (ie: 34.5 and 71.7) refer to?

9. Line 124: “but Korean students showed a large difference” the study is not designed to show a difference. Moreover p values are statistically significative in both study: I think that this important point deserve clarification

10. In table 2, in spite of the impossibility of statistical comparison, there is a marked difference between France and Korea in exposure to marketing strategies: do you have any explanation (within the discussion part)?

11. The data within tables 4 and 5 are original and were not in the study of Etain and al: perhaps it deserves more comments in the discussion?

12. Line 165: this sentence should be moved from the results section to the discussion and probably deserve a larger analysis regarding this paradoxical result.

13. Line 190: this sentence is unclear

Reviewer #2: Korean medical student attitudes on COI

PlosOne

Synopsis

This paper replicates a French study looking at attitudes and perceptions amongst medical students around conflicts of interest. It provides results of a survey of students and junior doctors about their reported understanding and views about conflict of interest and their experiences interacting with pharmaceutical industry representatives.

Overall comment

This is a useful paper and provides important information to document knowledge and attitudes about COI amongst Korean medical students/junior doctors. As far as I am aware this is the first paper to report on this topic in Korea. I commend the authors for reporting on this and look forward to seeing it add to the literature. We need these kinds of baseline papers in order to assess interventions to reduce the influence from industry on healthcare. I have some minor comments about wording and presentation, I hope the authors can address these without too much difficulty, to make this a more readable paper. I also suggest the authors are more circumspect in their claims about the degree to which their survey response is representative of the target population since they do not know the numbers in their target audience.

I did not receive any supplementary files to review. It would be useful for the reader to have access to the survey questions.

Specific comments:

INTRODUCTION

P3, line 39 / para 1 I do not understand the concept of ‘rebates in pharmaceutical sales’. Can the authors elaborate on the current system in Korea to which this refers? What are the rebates / who gets them / who gives them / how are they awarded etc.

P4, line 50 onwards / para 1 The authors’ representation of Etain et al’s paper was confusing. I think the syntax needs review. For example, it reads as if Etain et al were exposed to pharma reps a lot and had contradictory ideas about bias. I think the authors mean to say that Etain et al found that medical students were exposed etc.

P4, line 55 / para 3 The authors call this a qualitative survey (also on p 16) but it is a quantitative survey. They also say they compare attitudes amongst students, but they also compare knowledge, education, experiences with pharma reps etc.

P4, lines 59-63 / para 4 I am not sure this paper needs an ‘hypothesis’ approach. This paragraph could be reworded to simply present the aim.

Overall the introduction could benefit from some more intensive introductory discussion about COI amongst medical students / doctors – eg why this topic is important and worth studying. do eg explain why we care about it, what are the potential harms, what is the evidence of influence, what do we even mean by the term COI. There is much useful evidence on this topic that the authors could make use of – for example the authors might like to explore the reference list in a previous publication of mine on this topic

PARKER L (2019) I’m more susceptible to drug company money than I’d like to be. British Medical Journal, 19 December. Available at: https://blogs.bmj.com/bmj/2019/12/12/lisa-parker-im-more-susceptible-to-drug-company-money-that-id-like-to-be/

METHODS

The authors recruit students and junior doctors, but throughout the paper typically use the term ‘students’. I suggest adding a line to explain that the term student will be used to mean all the participants unless otherwise specified, so the readers are clear.

P5, line 80-81 / para 4 The authors do not make any kind of claim that they wrote to all medical schools or all resident institutions. They do not know the exact number who received the survey so cannot supply a response rate. Given all this, I suggest there is no substance to the claim they make in the Discussion that this represents the views of all Korean students. Nevertheless, their survey response numbers are substantial and this remains a very worthy study. I suggest the authors try to give some kind of context to their numbers to help readers who are unfamiliar with Korea – eg, how many medical schools are there in Korea, and can they estimate approximately how many residents there are (eg maybe the Medical Registration Board would assist?) or if those figures are unavailable, maybe estimate how many resident institutions there are?

RESULTS

P6, line 94 / para 3 The authors write ‘although [students] expressed interest in whether the courses they attended concerned COI …’ I am not sure what this means – can the authors expand?

P6, lines 96-100 / para 4 This paragraph lists ratios of items, I don’t understand what this means, what is the ratio the authors are referring to? Do they just mean % of students who answered ‘yes’ to each question?

P8, line 114 / para 1 The authors report that ‘35.5% in Set 1 and 45.6% in Set 2 thought monetary offerings generated a COI’. A similar statement is repeated on p9,line 120 / para 1 and on p 13, line 152/para 1. I don’t understand this statement. I thought Set 1 and Set 2 were just groupings of questions, not groupings of participants?

P11, line 142 / para 2 I suggest EDITING to read: “determine whether personal research on COI or lecture participation in COI affected BELIEFS ABOUT IMPACT OF COI ON future prescriptions BY SELF OR OTHERS more.”

Table 3 I found this table very hard to interpret, and the summary sentence after it (beginning “the analysis found that the P-value…”) was also hard to understand. Can the authors please clarify the meaning of this table, perhaps providing the results in a different format would help?

Table 4 I suggest the authors specify “Korean medical students” in the title, and can then delete the row of K K K K. This could be repeated in Table 5. I don’t understand the point of providing a P-value in this table or Table 5. I can’t see that it is relevant whether or not there is statistical significance between the ‘yes/no/don’t know’ groups.

P 13, line 158 “The degree to which it affects…” Can the authors explain what they are talking about here?

P14, line 167 / para 3 I suggest the following EDITS: “Regarding the STUDENTS’ PERCEPTION ABOUT possibility of bias…”

DISCUSSION

P 15, line 179 / para 4 I suggest omitting this presentation of the paper as an hypothesis approach, as mentioned earlier.

P 15, line 182 / parka 5 Can the authors clarify their meaning: “We met the minimum study population of 95% confidence level …”

P 16, line 185-188 As mentioned above, I don’t think the authors can claim that this survey represents the opinions of Korean medical students

P 16, line 200 / para 5 The authors suggest students should spend more time on quality education about COI, but I wonder whether this is ignoring the responsibility of the educational institutions to encourage accurate and interesting education for students. Can the authors provide any evidence / references about this.

P 17, line 17 / para 1 I really would like to see some stronger referencing of the literature on COI and suggests about systemic interventions that might reduce the influence of pharma reps / industry on students. This should go beyond individual disclosure / mandated transparency and might include topics such as : hospital / university policies to reduce pharma rep access to students, hospital / industry policies to limit the type of pharma rep gift giving and the amounts involved, university and hospital policies to limit the use of teachers who accept pharma industry gifts etc etc. Again, I encourage the authors to read widely around the topic of COI and how to reduce it so that they can provide a stronger argument and more useful suggestions for how Korea might address this issue in its healthcare system.

6. PLOS authors have the option to publish the peer review history of their article (what does this mean?). If published, this will include your full peer review and any attached files.

Reviewer #1: No

Reviewer #2: **Yes: **Lisa Parker

---

## [Author Response · Author response to Decision Letter 0]

23 Jul 2023

Response to Reviewers

Reviewers' comments:

Reviewer #1: 

Thank you very much for your study

I have some comments that I list in chronological order

Thank you so much

1. Line 27: “lecture participation in COI affected future prescriptions more” is unclear

Given the modifications made to the contents and format of Table 3, the corresponding comment section has been updated as follows.:

“As students moved up to higher education courses, they had greater interaction with pharmaceutical industry representatives, received more gifts, and displayed an increased likelihood of acknowledging the influence of COI on biases.” [Revised Manuscript Line 29-32] 

2. Line 50: “he was exposed to pharmaceutical sales representatives a lot, and he had a contradictory idea that he would not be biased, although others could be” is unclear: one could understand that Etain himself is exposed to pharmaceutical sales representatives while its conclusion concerns students.

We revised the sentence as follows: 

“In addition, many students were exposed to pharmaceutical representatives, and they held the contradictory belief that biases could affect others but not themselves.” [Revised Manuscript Line 86-88] 

3. Line 70: “a questionnaire was distributed to 10 students in advance to evaluate their 70 understanding and reflect their opinion on the subject”. Did you modify the questionnaire according to this first evaluation?

First of all, “70” is a number that means line number, and the subjects of the first evaluation were 10 students.

In addition, the following Korean expressions were clearly incorporate some of the opinions from the initial evaluation. For instance, there were suggestions to explicitly mention the gift or financial support provider (pharmaceutical laboratory) and to convey the intended meaning of the phrase ('impression of “being bought"') more clearly. 

4. Line 79: “The potential participants » : can you precise how many potentials participants were expected?

"The potential partners" means the total sample size of this study.

The number of subjects in this study is shown in the following table.

(* For detailed information, please see the attached file "1. Response to Reviewers")

The number of subjects in the above is not arbitrarily determined by the author but is based on data released by relevant public institutions.

5. Line 93: “Although they expressed interest in whether the courses they attended concerned COI (54.6%), about half said the professors teaching the courses did not disclose their COI information (46.4%)”: I think that these ideas are not related and should not be in the same sentence.

The questionnaire corresponding to the question is as follows.

4.6. I would like to know the COI of my teachers when they teach me?

 1) Yes (54.6%) 2) No 3) Don't know 

4.5. Do your teachers mention their COI during their lessons?

 1) Don't know 2) No (46.4%) 3) Some 4) All

We agree with you.

While the question holds individual value, I realized that it is inappropriate to express it in the same sentence since 'professor' and 'course' are not specified.

We revised the sentence as follows: 

“A large number of students reported insufficient education on COI (63.7%) and reported infrequent attendance in lectures (8.5%) or personal research (10.3%). Additionally, over half of the respondents (54.6%) expressed interest in COI information regarding their courses.” [Revised Manuscript Line 131-134]

6. Lines 105 to 107: the 51 students are probably the same in the 2 part of the paragraph but can you precise?

The questionnaire corresponding to the question is as follows.

3.2. During your studies, do you feel that you received enough information about declaration of COI?

 1) Yes 2) No 3) Don't know

3.3. During your studies, did you receive a lecture or a tutorial on COI?

 1) Yes 2) No 

2.1. Do you think you know how to define what a COI is?

 1) Yes 2) No 3) I'm not sure

There were 35 respondents who answered 'Yes' to question 3.2, 33 respondents who answered 'Yes' to question 3.3, and 17 respondents who answered 'Yes' to both questions. 

Hence, there were 51 unique respondents who answered 'Yes' after excluding duplicate responses (35+33-17=51). Additionally, out of these 51 respondents, 23 answered 'No' or 'Don't Know Actually' to question 2.1. 

Therefore, We believe the analysis result is accurate.

7. Table 1: can you precise that Etain and al wanted to investigate the Medical Students’ Knowledge of Situations at Risk of COI according to their level within medical school (and so did you)?

The previous research paper by Etain et al. describes the analysis results based on the curriculum level. The author of this paper also intended to conduct a comparative study using the findings from previous studies, so We think so.

8. Line 120 and 121: can you precise what this percentages (ie: 34.5 and 71.7) refer to?

The average score for Set 1 among French students is 34.5% [(27.7+32.9+35.0+35.6+41.5)/5=34.5%]. This value was found to be similar to the average score for Set 1 among Korean students [(30.9+39.7+45.1+26.5+35.1)/5=35.5%]

The average score for Set 2 among French students is 71.7% [(56.0+58.7+69.2+85.5+89.1)/5=71.7%]. It was observed that there is a significant difference from the average score for Set 2 among Korean students [(43.8+49.2+25.8+41.8+67.3)/5=45.6%]

9. Line 124: “but Korean students showed a large difference” the study is not designed to show a difference. Moreover p values are statistically significative in both study: I think that this important point deserve clarification

We agree with you

Therefore, We intend to address these points in the Discussion section.

“While each study conducted in France and Korea was not specifically designed to demonstrate differences, it would be meaningful to compare the research findings between the two countries using a consistent approach to validate any potential disparities.” [Revised Manuscript Line 232-234]

10. In table 2, in spite of the impossibility of statistical comparison, there is a marked difference between France and Korea in exposure to marketing strategies: do you have any explanation (within the discussion part)?

As with the response to the previous comment (9. Line 124), We believe it is meaningful to compare the results of each study conducted in France and Korea using a similar approach to identify any differences. 

Therefore, We intend to address these comments in the Discussion section.

11. The data within tables 4 and 5 are original and were not in the study of Etain and al: perhaps it deserves more comments in the discussion?

We believe that the knowledge about COI is related to the importance of education. Therefore, even though it was not included in the previous study, we have included it as an additional aspect.

However, not only does the format match that of Tables 1 and 2, but there is no issue with conveying the meaning through a description instead of using tables. Therefore, Tables 4 and 5 will be omitted, and their contents will be discussed in the following section.

“Further analysis revealed that respondents who believed they were knowledgeable about COI (50.1%) showed a greater awareness of COI compared to those who believed they lacked knowledge (31.8%). Additionally, they exhibited a higher awareness of direct COI (56.0%) compared to indirect COI (44.1%), as shown in Table 2.” [Revised Manuscript Line 216-219]

12. Line 165: this sentence should be moved from the results section to the discussion and probably deserve a larger analysis regarding this paradoxical result.

We agree with you.

We will move this sentence to the Discussion section and mention that further analysis is needed as a follow-up study.

“Further analysis revealed that respondents who believed they had knowledge about COI (18.9%) reported a higher incidence of receiving gifts compared to those who believed they lacked knowledge (3.0%). On the other hand, these results suggest that students with more exposure to pharmaceutical salespeople or receiving gifts had a better understanding of COI. Therefore, further analysis through follow-up studies is necessary.” [Revised Manuscript Line 254-258]

13. Line 190: this sentence is unclear

We revised the sentence as follows: 

“and the residents evenly participate in 18 out of the 26 training departments operating in Korea. With the exception of the 5 training departments with a total quota of less than 50, almost all departments are involved.” [Revised Manuscript Line 246-248]

 

Reviewer #2: 

PlosOne

Synopsis

This paper replicates a French study looking at attitudes and perceptions amongst medical students around conflicts of interest. It provides results of a survey of students and junior doctors about their reported understanding and views about conflict of interest and their experiences interacting with pharmaceutical industry representatives.

Overall comment

This is a useful paper and provides important information to document knowledge and attitudes about COI amongst Korean medical students/junior doctors. As far as I am aware this is the first paper to report on this topic in Korea. I commend the authors for reporting on this and look forward to seeing it add to the literature. We need these kinds of baseline papers in order to assess interventions to reduce the influence from industry on healthcare. I have some minor comments about wording and presentation, I hope the authors can address these without too much difficulty, to make this a more readable paper. I also suggest the authors are more circumspect in their claims about the degree to which their survey response is representative of the target population since they do not know the numbers in their target audience. 

I did not receive any supplementary files to review. It would be useful for the reader to have access to the survey questions. 

Thanks for your comment.

When We submit the 'Response to Reviewers' file, We will send you the 'Questionnaire' file together.

Specific comments:

INTRODUCTION

(P3, line 39 / para 1) I do not understand the concept of ‘rebates in pharmaceutical sales’. Can the authors elaborate on the current system in Korea to which this refers? What are the rebates / who gets them / who gives them / how are they awarded etc.

We revised it by adding the description as below :

“In general, rebates refer to the act or the amount by which a seller reduces the selling price by refunding a portion of the sale to the buyer to promote sales.

However, under the Korean Pharmaceutical Affairs Act, the pharmaceutical sales rebate refers to a 'kickback' wherein a doctor receives private profits in correspondence to the prescription and purchase of necessary drugs from a pharmaceutical company or supplier while prescribing drugs for medical treatment.

Types of pharmaceutical sales rebates announced by Korea's Health Insurance Review and Assessment Service (HIRA) include ① drug adoption case fees, ② drug prescription case fees, ③ regular advisory fees, ④ tourism or meal expenses, and ⑤ expenses related to overseas conference attendance.

According to the Ministry of Health and Welfare's 'Notification Status of Illegal Rebate Detection between 2015 and 2018,' it appears that illegal rebates have been gradually decreasing. However, it has been discovered that rebates are not actually decreasing, but rather that pharmaceutical companies are providing rebates through sales agencies known as CSOs (Contract Sales Organizations), resulting in their exclusion from the statistics. They are taking advantage of the fact that CSOs are not classified as drug suppliers under the Pharmaceutical Affairs Act, making it impossible to impose penalties even if they are caught.(1)” [Revised Manuscript Line 57-73]

(P4, line 50 onwards / para 1) The authors’ representation of Etain et al’s paper was confusing. I think the syntax needs review. For example, it reads as if Etain et al were exposed to pharma reps a lot and had contradictory ideas about bias. I think the authors mean to say that Etain et al found that medical students were exposed etc. 

We revised the sentence as follows: 

“In addition, many students were exposed to pharmaceutical representatives, and they held the contradictory belief that biases could affect others, but not themselves.” [Revised Manuscript Line 86-88] 

(P4, line 55 / para 3) The authors call this a qualitative survey (also on p 16) but it is a quantitative survey. They also say they compare attitudes amongst students, but they also compare knowledge, education, experiences with pharma reps etc. 

We revised the sentence as follows: 

“In this context, we also conducted a survey of medical students in Korea to collect quantitative information on knowledge, education, personal exposure, and opinions about COI and to identify priorities for future education on COI.” [Revised Manuscript Line 92]

“and (2) it provides quantitative data on potential differences in perception according to curriculum stage, including among preclinical and clinical students.” [Revised Manuscript Line 252]

(P4, lines 59-63 / para 4) I am not sure this paper needs an ‘hypothesis’ approach. This paragraph could be reworded to simply present the aim. 

We decided to understand and accept your opinion, and revised the sentence as follows:

“This paper aims to identify the following. : “ [Revised Manuscript Line 95-96]

Overall the introduction could benefit from some more intensive introductory discussion about COI amongst medical students / doctors – eg why this topic is important and worth studying. do eg explain why we care about it, what are the potential harms, what is the evidence of influence, what do we even mean by the term COI. 

There is much useful evidence on this topic that the authors could make use of – for example the authors might like to explore the reference list in a previous publication of mine on this topic 

PARKER L (2019) I’m more susceptible to drug company money than I’d like to be. British Medical Journal, 19 December. Available at: https://blogs.bmj.com/bmj/2019/12/12/lisa-parker-im-more-susceptible-to-drug-company-money-that-id-like-to-be/

We added a description of COI in the introduction section as below.

“There are many studies (2) (3) demonstrating the impact of pharmaceutical companies' marketing activities on doctors' prescribing practices. Consequently, pharmaceutical companies are compelled to engage in a range of marketing activities, both direct and indirect, in order to influence doctors' prescriptions.

Students are appealing marketing targets for pharmaceutical companies because established prescribing habits are not easily altered. As a result, medical students and junior doctors are susceptible to being exposed to conflicts of interest with pharmaceutical companies. (4)

A conflict of interest is a set of circumstances that creates a risk that professional judgment or actions regarding a primary interest(ex. the welfare of patients) will be unduly influenced by a secondary interest(ex. financial gain). (5)

Conflict of interest is a specific 'circumstance.' Therefore, while it may not be a problem in itself, it has a high likelihood of leading to biases. Hence, proactive measures are necessary to prevent it, and "disclosure" is widely recognized as a common solution. (6)” [Revised Manuscript Line 39-52]

METHODS

The authors recruit students and junior doctors, but throughout the paper typically use the term ‘students’. I suggest adding a line to explain that the term student will be used to mean all the participants unless otherwise specified, so the readers are clear. 

We revised the sentence as follows

: In total, 388 medical students, interns and residents (hereafter referred to as ”students”). [Revised Manuscript Line 23]

Furthermore, the term 'medical students, interns, and residents' was consolidated into 'students' for consistency.

(P5, line 80-81 / para 4) The authors do not make any kind of claim that they wrote to all medical schools or all resident institutions. They do not know the exact number who received the survey so cannot supply a response rate. Given all this, I suggest there is no substance to the claim they make in the Discussion that this represents the views of all Korean students. Nevertheless, their survey response numbers are substantial and this remains a very worthy study. I suggest the authors try to give some kind of context to their numbers to help readers who are unfamiliar with Korea– eg, how many medical schools are there in Korea, and can they estimate approximately how many residents there are (eg maybe the Medical Registration Board would assist?) or if those figures are unavailable, maybe estimate how many resident institutions there are? 

Although the paper did not mention the minimum number of subjects, the investigation included the number of medical students and residents to calculate the minimum number of subjects during the establishment of the research plan. The main contents are as follows.

(* For detailed information, please see the attached file "1. Response to Reviewers")

RESULTS

(P6, line 94 / para 3) The authors write ‘although [students] expressed interest in whether the courses they attended concerned COI …’ I am not sure what this means – can the authors expand? 

The author did not intend to arbitrarily expand the meaning, but rather based it on the analysis results of the survey response status.

The questionnaire corresponding to the question is as follows.

4.6. I would like to know the COI of my teachers when they teach me?

 1) Yes (54.6%) 2) No 3) Don't know 

4.5. Do your teachers mention their COI during their lessons?

 1) Don't know 2) No (46.4%) 3) Some 4) All

Although it is valuable as an individual question, I realized that it is inappropriate to express it in the same sentence because 'professor' and 'courses' are not specified.

We revised the sentence as follows: 

“A large number of students reported insufficient education on COI (63.7%), rarely attended lectures (8.5%), or conducted personal research (10.3%). Additionally, more than half (54.6%) of the respondents expressed interest in COI information about their courses.” [Revised Manuscript Line 131-134]

(P6, lines 96-100 / para 4) This paragraph lists ratios of items, I don’t understand what this means, what is the ratio the authors are referring to? Do they just mean % of students who answered ‘yes’ to each question? 

To provide a clear depiction of the response rate to the questionnaire, we have revised the sentence and included a table for better understanding.

“Upon comparing the results of previous studies conducted in France and Korea, the proportions of the three items(3.3./3.4./4.6.) were similar, as depicted in the table 1 below. However, it was notable that French students expressed a significantly higher level of dissatisfaction regarding the adequacy of COI information(3.2./4.5).” [Revised Manuscript Line 139-143]

(* For detailed information, please see the attached file "1. Response to Reviewers")

(P8, line 114 / para 1) The authors report that ‘35.5% in Set 1 and 45.6% in Set 2 thought monetary offerings generated a COI’. A similar statement is repeated on p9,line 120 / para 1 and on p 13, line 152/para 1. I don’t understand this statement. I thought Set 1 and Set 2 were just groupings of questions, not groupings of participants? 

As the reviewer mentioned, Set1 and Set2 represent groups of questions. However, there was an error in expressing the analysis result.

We revised the sentence as follows: 

“Give this, 35.5% in Set 1 and 45.6% in Set 2 considered monetary offerings as COI, demonstrating a higher awareness of direct offerings than indirect offerings.” [Revised Manuscript Line 159-161]

“Compared to the results of previous studies in France, Set 1 (34.5%) was at a similar level to Korea, but Set 2 (71.7%) showed a significantly higher level of awareness than Korean students.” [Revised Manuscript Line 166]

(P11, line 142 / para 2) I suggest EDITING to read: “determine whether personal research on COI or lecture participation in COI affected BELIEFS ABOUT IMPACT OF COI ON future prescriptions BY SELF OR OTHERS more.”

(Table 3) I found this table very hard to interpret, and the summary sentence after it (beginning “the analysis found that the P-value…”) was also hard to understand. Can the authors please clarify the meaning of this table, perhaps providing the results in a different format would help?

First, reflecting your opinion, we have made some changes to the format of Table 3 as shown below.

Beta is the data used to calculate the OR(Odds Ratio) and CI(Confidence Interval), and SD (Standard Deviation) represents the variability of the response. Therefore, these two factors are omitted as they do not hold significant meaning in interpreting the analysis results.

Furthermore, data points with p-values above the significance level (<0.05) are excluded, as they lack statistical significance.

Second, I have added the following explanation

(* For detailed information, please see the attached file "1. Response to Reviewers")

“Table 3 demonstrates a 95% confidence interval indicating that for each increment in the class level, the likelihood of answering 'Yes' to that question increases by a minimum of 1.95 times and a maximum of 3.33 times.” [Revised Manuscript Line 194-197]

(Table 4) I suggest the authors specify “Korean medical students” in the title, and can then delete the row of K K K K. This could be repeated in Table 5. I don’t understand the point of providing a P-value in this table or Table 5. I can’t see that it is relevant whether or not there is statistical significance between the ‘yes/no/don’t know’ groups. 

(P 13, line 158) “The degree to which it affects…” Can the authors explain what they are talking about here? 

(P14, line 167 / para 3) I suggest the following EDITS: “Regarding the STUDENTS’ PERCEPTION ABOUT possibility of bias…” 

We believe that the knowledge about COI is related to the importance of education. Therefore, even though it was not included in the previous study, we have included it as an additional aspect.

However, not only does the format match that of Tables 1 and 2, but there is no issue with conveying the meaning through a description instead of using tables. Therefore, Tables 4 and 5 will be omitted, and their contents will be discussed in the following section.

“Further analysis revealed that respondents who believed they had knowledge about COI (50.1%) exhibited a greater awareness of COI compared to those who believed they lacked knowledge (31.8%). Additionally, they demonstrated a higher awareness of direct COI (56.0%) compared to indirect COI (44.1%), as indicated in Table 1.” [Revised Manuscript Line 216-219]

DISCUSSION

(P 15, line 179 / para 4) I suggest omitting this presentation of the paper as an hypothesis approach, as mentioned earlier. 

We have decided to understand and accept your opinion, so we omitted the sentence. [Revised Manuscript Line 235-236]

(P 15, line 182 / para 5) Can the authors clarify their meaning: “We met the minimum study population of 95% confidence level …” 

As the answer to the question (P5, line 80-81 / para 4) above, the minimum number of participants was calculated with a confidence level of 95% and a margin of error of 5%, and the result was 380.

The number of participants in this study is 388, which meets this condition.

(P 16, line 185-188) As mentioned above, I don’t think the authors can claim that this survey represents the opinions of Korean medical students. 

As the answer to the question (P5, line 80-81 / para 4) above, the minimum number of participants was calculated with a confidence level of 95% and a margin of error of 5%, and the result was 380.

The number of participants in this study is 388, which meets this condition.

As described in (P5, line 181-185) above, there is a limitation regarding participant concentration in specific areas and stages. However, despite these limitations, we believe the study still provides representation.

(P 16, line 200 / para 5) The authors suggest students should spend more time on quality education about COI, but I wonder whether this is ignoring the responsibility of the educational institutions to encourage accurate and interesting education for students. Can the authors provide any evidence / references about this. 

We have included additional information regarding the challenges of medical ethics education for medical students in Korea, along with a relevant paper as a reference. 

“Currently, medical ethics education for medical students in Korea faces challenges due to an absolute lack of class time, insufficient presence of related scholars, unclear educational goals, and inadequate development of methodologies. Additionally, there is a lack of education specifically addressing COI (7) (8)” [Revised Manuscript Line 263-266]

(P 17, line 17 / para 1) I really would like to see some stronger referencing of the literature on COI and suggests about systemic interventions that might reduce the influence of pharma reps / industry on students. This should go beyond individual disclosure / mandated transparency and might include topics such as : hospital / university policies to reduce pharma rep access to students, hospital / industry policies to limit the type of pharma rep gift giving and the amounts involved, university and hospital policies to limit the use of teachers who accept pharma industry gifts etc etc. Again, I encourage the authors to read widely around the topic of COI and how to reduce it so that they can provide a stronger argument and more useful suggestions for how Korea might address this issue in its healthcare system. 

Thank you for your suggestion and encouragement.

We deeply sympathize with the need for policies to reduce the impact of the pharmaceutical industry on medical students. Understanding the nature of COI and preparing policies to reduce biases caused by COI will be reflected in further follow-up studies.

Thank you again for your suggestion and encouragement.

 

1. JO S-G. A Study on the Rebate in the Medical Industry. Seoul Korea: Hanyang University Law School; 2020.

2. Wazana A. Physicians and the Pharmaceutical IndustryIs a Gift Ever Just a Gift? JAMA. 2000;283(3):373-80.

3. DeJong C, Aguilar T, Tseng CW, Lin GA, Boscardin WJ, Dudley RA. Pharmaceutical Industry-Sponsored Meals and Physician Prescribing Patterns for Medicare Beneficiaries. JAMA Intern Med. 2016;176(8):1114-22.

4. CHEONG Yoo-Seok PJ-H, Younsuck Koh. Ethical Issues Concerning the Relationship between Medical Students/Residents and the Pharmaceutical Industry. Korean J Med Ethics. 2011;14(2):215-23.

5. Thompson DF. Understanding financial conflicts of interest. N Engl J Med. 1993;329(8):573-6.

6. Medicine Io. Conflict of interest in medical research, education, and practice. Washington, D.C.: National Academies Press; 2009.

7. Choi K. Medical Ethics and Professional Education: Educational Status and Roles of Philosophy. Journal of human studies. 2007;-(12):218-43.

8. KWON I. A Critical Review of the Current Medical Ethics Education in Korea. Korean Journal of Medical Ethics. 2006;9(1):60-72.

---

## [Decision Letter · Decision Letter 1]

24 Aug 2023

PONE-D-23-08482R1Attitudes of medical students on conflict of interest: A comparative study of Korea and FrancePLOS ONE

Dear Dr. Choe,

Thank you for submitting your manuscript to PLOS ONE. After careful consideration, we feel that it has merit but does not fully meet PLOS ONE’s publication criteria as it currently stands. Therefore, we invite you to submit a revised version of the manuscript that addresses the points raised during the review process. One reviewer has made several comments and suggestions for improvement. I invite you to take advantage of this opportunity to further improve your manuscript before it is potentially accepted for publication. Please submit your revised manuscript by Oct 08 2023 11:59PM. If you will need more time than this to complete your revisions, please reply to this message or contact the journal office at plosone@plos.org. Please include the following items when submitting your revised manuscript:A rebuttal letter that responds to each point raised by the academic editor and reviewer(s). You should upload this letter as a separate file labeled 'Response to Reviewers'.A marked-up copy of your manuscript that highlights changes made to the original version. You should upload this as a separate file labeled 'Revised Manuscript with Track Changes'.An unmarked version of your revised paper without tracked changes. You should upload this as a separate file labeled 'Manuscript'.If applicable, we recommend that you deposit your laboratory protocols in protocols.io to enhance the reproducibility of your results. Protocols.io assigns your protocol its own identifier (DOI) so that it can be cited independently in the future. For instructions see: https://journals.plos.org/plosone/s/submission-guidelines#loc-laboratory-protocols. Additionally, PLOS ONE offers an option for publishing peer-reviewed Lab Protocol articles, which describe protocols hosted on protocols.io. Read more information on sharing protocols at https://plos.org/protocols?utm_medium=editorial-email&utm_source=authorletters&utm_campaign=protocols.

We look forward to receiving your revised manuscript.

Kind regards,

Alberto Molina Pérez, Ph.D.

Academic Editor

PLOS ONE

Journal Requirements:

Reviewers' comments:

Reviewer's Responses to Questions

**Comments to the Author**

1. If the authors have adequately addressed your comments raised in a previous round of review and you feel that this manuscript is now acceptable for publication, you may indicate that here to bypass the “Comments to the Author” section, enter your conflict of interest statement in the “Confidential to Editor” section, and submit your "Accept" recommendation.

Reviewer #2: (No Response)

2. Is the manuscript technically sound, and do the data support the conclusions?

Reviewer #2: Partly

3. Has the statistical analysis been performed appropriately and rigorously? 

Reviewer #2: I Don't Know

4. Have the authors made all data underlying the findings in their manuscript fully available?

Reviewer #2: No

5. Is the manuscript presented in an intelligible fashion and written in standard English?

Reviewer #2: Yes

6. Review Comments to the Author

Reviewer #2: The authors have addressed many of my comments. There are, however, still some omissions in the text and some sentences and Results Tables where the meaning is not clear. The Abstract needs more work. Further revision should be relatively straight forward.

Abstract: This needs more work to act as a succinct representation of the paper. I suggest the first sentence should focus more on the problem – eg use line 43 in the paper: ‘Medical students are appealing marketing targets for pharmaceutical companies’ and then continue from line 19 about Etain’s study, and your replication of it. The background on Korean law (lines 16-19) is better kept in the paper alone. I would like a clear, succinct description of the method eg [1] an empirical survey study with Korean medical students concerning their educational experiences and views on conflicts of interest and [2] comparing and contrasting the results with Etain’s study of French medical students. The current method description (line 22-23) is very confusing – it suggests the authors are carrying out a review of ‘previous studies in Korea’ which is not correct. I found the wording of results in the Abstract (lines 24-25) was hard to follow. The authors might consider a revision such as: ‘Receipt of direct or indirect financial offerings from pharmaceutical industries was not properly recognised as COI by students.’ Finally, the last sentence in the Abstract should stick to the third person grammar and avoid using a second person approach like ‘your own’ (line 35).

Introduction

Line 40 I suggest pharma companies are not ‘compelled to’ engage in marketing, this word could be omitted

Line 43 The authors should repeat here the phrasing in the Abstract line 23 about ‘students, interns and residents, hereafter referred to as students’ – writing it in the Abstract alone is not sufficient

Line 52 I strongly disagree that disclosure is a solution for preventing conflicts of interest, but I recognise it is often reported this way in the literature. The authors could consider a comment about the controversy around this – eg others (myself included) regard disclosure as a necessary first step, although not sufficient for preventing COI. I expanded on this and other possible solutions in my previous review and the authors might find this reference helpful

Parker L, Karanges EA, Bero L. Changes in the type and amount of spending disclosed by Australian pharmaceutical companies: an observational study. BMJ Open 2019;9:e024928. doi:10.1136/ bmjopen-2018-024928

Line 64 What is the difference between drug adoption case fees and drug prescription case fees?

Line 67 The paragraph here would be better moved to AFTER the paragraph at line 74. This would enable better flow of subjects in the text.

Line 96 This was confusing. It needs to be re-worded as aims, not as hypotheses. I suggest editing to something like this (my suggested edits in capitals) : "[1] Korean students’ ABILITY TO identify some common situations as a potential COI; [2] Korean students REPORTS AND VIEWS ABOUT their education on COI; [3] Korean students PERCEPTIONS OF the bias induced by COI ON THEMSELVES AND OTHERS"

Results

Line 141 The authors report French and Korean student answers to item 4.6 are similar, but Table 1 shows they are quite different (66.6 % vs 54.6%)

Line 143 student dissatisfaction with COI information is more properly shown in item 4.6 than 4.5

Table 1, I don’t understand what ‘gab’ refers to (head in final column)

Line 146-147 I don’t understand what this sentence is about, can the authors reword to clarify.

Line 150 Do the authors mean to say that 51 students answered yes to BOTH questions, or to at least one of the questions listed? And when they say 23 out of 51 respondents declared they cannot define COI, is this the same 51 who answered yes to both (or at least one of?) the questions listed?

Table 2 and 3 I still don’t understand the p value. Is it reporting homogeneity between preclinical, clinical and residents within one country, or is it reporting homogeneity between preclinical in France and Korea; clinical in France and Korea, and residents in France and Korea? Neither seem to be feasible, given the variation in results so more information is needed about the figures in order to convince me that they are correct.

Line 177 It seems that Table 4 and associated text should be put here

Line 183 I don’t understand this comment, please edit to clarify

Line 218 Who are ‘they’?

Line 220 I don’t understand this comment, please edit to clarify

Discussion

Line 232 can the authors provide a 1-2 line summary of the comparative results here?

Line 237 onwards. The limitations section needs work. The comment about getting to statistical significance is not relevant in relation to % of the student population surveyed. The study has plenty of merit so I suggest editing this section, starting with the study strengths (diversity of participants at all levels – students, interns, residents; even participation from a broad range of training departments ie 18/26) and then acknowledge study limitations (the majority being clinical; most concentrated in just a few regions). As previously mentioned in my original review, the authors cannot assume that the study represents the opinions of the majority of Korean medical students because they do not know the number of total students in the country. I would omit that paragraph starting 244.

Line 254 onwards This paragraph discussing the results would be better moved up earlier in the Discussion, after paragraph 3 of the Discussion (ie before the Limitations section)

Line 261 I don’t understand the comment/interpretation that this study shows students’ have a ‘basic sense of ethics.’ Can the authors edit to clarify or omit.

Line 267 Can this authors justify this paragraph / comment, eg with reference to the literature.

Line 270 As mentioned, I disagree that COI disclosure will prevent bias, but I acknowledge that what the authors have written is a widely held view.

7. PLOS authors have the option to publish the peer review history of their article (what does this mean?). If published, this will include your full peer review and any attached files.

Reviewer #2: **Yes: **Lisa Parker

---

## [Author Response · Author response to Decision Letter 1]

4 Oct 2023

Abstract: We revised the sentence as follows:

“Medical students are potential marketing targets for pharmaceutical companies because established prescribing habits are not easily altered. In 2014, Bruno Etain and several other researchers published a paper which investigated the knowledge of and opinions on potential conflict of interest (COI) with regard to preclinical and clinical students enrolled in medical schools in France and residents working in hospitals. An empirical survey study with Korean medical students concerning their educational experiences and views on conflicts of interest and comparing and contrasting the results with Etain’s study of French medical students. Receipt of direct or indirect financial offerings from pharmaceutical industries was not properly recognised as COI by the medical students. Therefore, strengthening education on COI and implementing institutional improvements for COI disclosure are essential to prevent bias caused by COI and enhance awareness levels regarding COI.” [3. Manuscript (Revision 2) Line 16-26]

Introduction

Line 40 We revised the sentence as follows: 

“For these reasons, pharmaceutical companies are conducting various direct and indirect marketing activities to influence doctors' prescriptions.” [3. Manuscript (Revision 2) Line 29-30]

Line 43 We revised the sentence as follows: 

“Medical students, interns and residents (hereafter referred to as “students”) are potential marketing targets for pharmaceutical companies because established prescribing habits are not easily altered.” [3. Manuscript (Revision 2) Line 31]

Line 52 Thank you for recommending the paper, and We have reviewed its contents in full.

As mentioned in the paper you recommended, regarding the importance of 'Transparency', 1) Transparency may assist those reading or receiving the disclosure to judge the risk of bias in those making the disclosure. 2) The act of disclosure may bring about changes where individuals refrain from engaging in related behaviors to avoid suspicions of bias arising. Therefore, in situations where voluntary disclosure is unlikely, We agree that measures are needed to compel the disclosure of conflicts of interest through the establishment of relevant laws and regulations, in order to prevent bias from occurring.

However, we aimed to approach conflicts of interest from an professional ethics perspective. Conflicts of interest are considered to be an moral issue for each individual, an professional ethics issue, and an issue related to human nature. Therefore, it can be relative depending on the differences in the individual's circumstances (e.g., income level, education level etc.) and their values (e.g., prioritizing professional integrity over financial benefits etc.).

Ethical issues cannot be perfectly controlled or resolved by regulatory means, as we believe that laws and policies are designed to restrict significant wrongdoings that exceed social standards and justice. The disclosure mentioned earlier in the document from the Institute of Medicine (IOM) is described as a "common solution" from this ethical perspective and appears to encompass a broader concept than disclosure from a legal and institutional standpoint. Therefore, in order to prevent bias in individuals facing conflicts of interest, there is a need for an ethical awareness of potentially risky situations, and various forms of education are essential to foster this correct awareness.

Line 64 

"drug adoption case fees" refers to case fees agreed upon for supplying pharmaceuticals to healthcare institutions and is also known as "landing fees."

"drug prescription case fees" refer to payments made in proportion to the prescription amount to induce the prescription of specific pharmaceuticals, and it is also referred to as "matching fees."

Line 67 We moved the paragraph [3. Manuscript (Revision 2) Line 58-64] 

Line 96 We revised the sentence as follows: 

“This paper aims to identify the following: ① Korean students’ ability to identify some common situations as a potential COI; ② Korean students reports and views about their education on COI; ③ Korean students perceptions of the bias induced by COI on themselves and others.” [3. Manuscript (Revision 2) Line 78-80]

Results

Line 141 We revised the sentence as follows:

“Upon comparing the results of previous studies conducted in France and Korea, the proportions of the two items(3.3./3.4.) were similar, as depicted in the table 1 below.” [3. Manuscript (Revision 2) Line 112] 

Line 143 We revised the sentence as follows:

“However, it was notable that French students expressed a significantly higher level of dissatisfaction regarding the adequacy of COI information(3.2./4.6.)” [3. Manuscript (Revision 2) Line 114]

We also used the term "gab" to refer to the difference between numbers compared in a statistical table, but We realized that it was not an appropriate expression. The term "gab" has been changed to "difference." [3. Manuscript (Revision 2) Line 115 Table 1]

Line 146-147 

Lines 146-147 aimed to convey the response to the question, "Do you think you know how to define what a COI is?", and the responses were as follows.

We revised the sentence as follows:

“In response to the question, "Do you think you know how to define what a COI is?", only 24.8% responded that they could define it, and there was no significant difference by class level (PC: 26.8%; C: 25.9%; R: 32.1%; P = 0.1385). Compared to the results of a previous study in which 64.6% of French students claimed they could define it, Korean students' knowledge about COI was found to be low (24.8%).” [3. Manuscript (Revision 2) Line 117-121]

Line 150 We revised the sentence as follows:

Students who answered "Yes" to at least one of the following two questions were 51 (13.1%): "Do you think you received enough information about COI disclosure?" and "Have you ever received a special lecture or individual training on COI?”. Furthermore, out of the 51 respondents who answered "Yes" to at least one of the questions, 23 (45.1%) of them responded with "No" or "I'm not sure" to the question "Do you think you know how to define what a COI is?", indicating that COI education is not being properly conducted. [3. Manuscript (Revision 2) Line 122-127]

Table 2 and 3 

We should consider the p-values for France and Korea independently in Table 2 and Table 3. In other words, it represents the homogeneity of responses based on class level in each case for France and Korea, respectively.

Additionally, it is expected that the reviewer interprets lower p-values in Table 2 as indicating higher homogeneity. (Hence the numerical discrepancy you mentioned.)

However, our conducted chi-square test has the following null and alternative hypotheses, and if the p-value is less than the significance level (0.05), we reject the null hypothesis (H0) and accept the alternative hypothesis (H1).

Null Hypothesis (H0): There is no difference in responses based on class level. 

Alternative Hypothesis (H1): There is a difference in responses based on class level.

In other words, the lower the p-value, the lower the homogeneity of responses, and the higher the p-value, the higher the homogeneity.

Therefore, if the p-value is below the significance level, we can interpret it as a statistically significant difference in responses based on class level.

Furthermore, in the case of Table 3, it is used to determine whether the logistic regression results are statistically meaningful, regardless of homogeneity.

Line 177 We moved the paragraph [3. Manuscript (Revision 2) Line 162-165]

Line 183 We revised the sentence as follows:

“In particular Table 3, the responses to the question "Have you ever received a gift from the PI?" showed the largest difference between Korean (15.7%) and French (62.7%) students, while the responses to the question "I consider it as a COI when attending a meal sponsored by the PI" showed the smallest difference between Korean (17.3%) and French (21.4%) students.” [3. Manuscript (Revision 2) Line 170-173]

Line 218 We moved the paragraph and revised the sentence as follows: 

“In the additional analysis of Table 2, we applied the responses to the question "Do you think you can define what a conflict of interest is?" instead of the responses to the question "What is your class level?" The results showed that respondents who answered "Yes" had a higher awareness of COI (50.1%) compared to those who answered "No" (31.8%). Furthermore, among the respondents who answered "Yes," the awareness of direct COI (Set 2) (56.0%) was higher than the awareness of indirect COI (Set 1) (44.1%). “ [3. Manuscript (Revision 2) Line 145-150].

Line 220 Further analysis of Table 3, we also examined the responses to the question "Do you think you can define what a conflict of interest is?" instead of the question "What is your class level?" As a result, the overall percentage of respondents who answered "Yes" (37.9%) and "No" (40.1%) was similar. Furthermore, among those who answered "Yes," the percentage of those who believe that others can develop bias (68.2%) was higher than the percentage of those who believe they can develop bias (14.2%). [3. Manuscript (Revision 2) Line 174-179].

Discussion

Line 232 We believe the summary of the requested comparative results is described in the paragraph just before Line 228. Therefore, We have moved Line 232 paragraph forward and made some modifications to its content. :

“Additionally, while each study conducted in France and Korea was not specifically designed to demonstrate differences, it would be meaningful to compare the research findings between the two countries using a consistent approach to validate any potential disparities.” [3. Manuscript (Revision 2) Line 182-185].

Line 237 We revised the sentence as follows: 

However, The main potential limitation of this study is the lack of representation, which is also showed in previous studies. The majority of respondents were locally concentrated in some regions (Seoul, Gyeonggi, Gangwon), with more than half of them being enrolled at the clinical level (PC: 14.4%; C: 64.6%; R: 20.8%). This imbalance is expected to reflect the following reasons: The survey on COI for students in Korea is the first attempt. In addition, many surveyed organizations were reluctant to distribute the questionnaire due to lack of understanding of COI and administrative burden.” [3. Manuscript (Revision 2) Line 199-204].

Line 254 We moved and revised the paragraph as follows : 

“The strengths of this study include the diverse participation of medical students, interns, and residents at all stages of medical education. In particular, residents evenly participated in 18 out of the 26 training departments operating in Korea, with the exception of 5 departments with a total quota of less than 50.” [3. Manuscript (Revision 2) Line 195-198].

Line 261 It is common to believe that under the same conditions of a COI situation, the likelihood of bias occurring is the same for oneself as it is for others. However, thinking that even though bias is likely to occur in others, the likelihood of bias occurring in oneself is low can be seen as an expression of one's determination not to go in the wrong direction. This is what we have referred to as a "basic sense of ethics."

Line 267 In Korea, not only is there a shortage of research literature on COI, but this is even more pronounced when it comes to education about COI. However, it is worth noting that, as an alternative to research literature, there are currently no courses dedicated to the topic of COI in the curriculum of Korean medical schools.

---

## [Editor Report · Decision Letter 2]

20 Oct 2023

Attitudes of medical students on conflict of interest: A comparative study of Korea and France

PONE-D-23-08482R2

Dear Dr. Choe,

We’re pleased to inform you that your manuscript has been judged scientifically suitable for publication and will be formally accepted for publication once it meets all outstanding technical requirements.

Kind regards,

Alberto Molina Pérez, Ph.D.

Academic Editor

PLOS ONE
---

## [Editor Report · Acceptance letter]

24 Oct 2023

PONE-D-23-08482R2 

Attitudes of medical students on conflict of interest: A comparative study of Korea and France 

Dear Dr. Choe:

I'm pleased to inform you that your manuscript has been deemed suitable for publication in PLOS ONE. Congratulations! Your manuscript is now with our production department. 

Kind regards, 

on behalf of

Dr. Alberto Molina Pérez 

Academic Editor

PLOS ONE